# Structural and Rheological Properties of Pectins Extracted from Industrial Sugar Beet By-Products

**DOI:** 10.3390/molecules24030392

**Published:** 2019-01-22

**Authors:** M. Teresa Pacheco, Mar Villamiel, Rodrigo Moreno, F. Javier Moreno

**Affiliations:** 1Instituto de Investigación en Ciencias de la Alimentación (CIAL) (CSIC-UAM) CEI (CSIC+UAM), Campus de la Universidad Autónoma de Madrid, Nicolás Cabrera 9, 28049 Madrid, Spain; mayte@cial.uam-csic.es (M.T.P.); javier.moreno@csic.es (F.J.M.); 2Instituto de Cerámica y Vidrio (ICV), Consejo Superior de Investigaciones Científicas (CSIC), 28049 Madrid, Spain; rmoreno@icv.csic.es

**Keywords:** sugar beet, by-products, silage, pectin, viscosity, emulsifying activity

## Abstract

In this work, the efficient extraction of pectin from sugar beet by-products (pressed, ensiled and dried pulp), by using an acid method or a commercial cellulose, is accomplished. The extraction method had an impact on the pectin monomeric composition, mainly in xylose, arabinose, and galacturonic acid content, as determined by GC-FID. FTIR and SEC analyses allowed the determination of similar degrees of methoxylation and molecular weights, respectively, in the extracted pectins. The acid extraction of pectin in the ensiled by-product led to the highest yield (19%) with a galacturonic acid content of 46%, whereas the application of the enzymatic extraction method resulted in a lower yield (13%) but higher galacturonic acid content (72%). Moreover, the stability in aqueous solution as well as the emulsifying activity index was higher for pectin extracted by the acid method, whereas the viscosity was higher in pectin extracted by the enzymatic method. To the best of our knowledge, this is the first study analyzing the physicochemical properties and exploring the potential reuse of ensiled and dried by-products from sugar beet industry for the extraction of pectin to be further used in the food and pharmaceutical areas.

## 1. Introduction

In 2016, the largest area of root crops (1.7 million hectares) in the European Union (EU) was occupied by potatoes closely followed by sugar beet (*Beta vulgaris L.* subsp. *vulgaris* var. altissima Döll) (1.5 million hectares) [1]. These values point out the EU as the leading producer of sugar beet, providing approximately 50% of the global production, whose process generates a volume waste of 111.6 million tons per year. In addition, in Spain, sugar beet is the only source of sugar, producing 3000 tons of residues per year.

Only 30% of the world’s sugar production comes from sugar beet, whereas the rest is derived from cane [1]; however, the obtainment of sugar from beet generates a significant volume of wastes each year, which is considered of great importance in terms of underexploited opportunities and generated levels [2]. When the sugar beet residues are exploited, habitually they are used as lignocellulosic material for the ethanol obtaining and pectin extraction [3].

Pectin, an important anionic heteropolysaccharide, exists in the cell walls of dicotyledonous plants [4], and over the last years, pectin has gained increasing interest as thickening or gelling agent for the chemical and food industry [5]. Furthermore, pectin has been described as an emerging prebiotic with the ability to modulate the bacterial composition of the colon microbiota [6], being able to exert beneficial effects on health.

Sugar beet pectin (SBP), compared to the main sources of pectin that are citrus and apple, has poorer gelling properties due to its higher content of neutral sugars, low presence of acetyl groups (4–5%) [7], content of ferulic acid, higher protein content [8], and/or its relatively low molecular mass, but in contrast, these molecular characteristics give the SBP better emulsifying properties [9,10].

However, depending on the applied extraction method, the structure and technological properties of SBP can widely vary. A large number of studies have addressed the extraction and properties of pectin from sugar beet pulp pressed (SBP-P) in recent years, and most of the studies have been focused on the effect of extractants and extraction conditions on pectin yield, chemical composition, and technological behavior [11,12,13]. However, there is an excess of other underutilized industrial sugar beet by-products, such as ensiled sugar beet (SBP-E) and dried sugar beet pulp (SBP-D) whose potential as raw materials for obtaining similar compounds has not yet been addressed.

Therefore, the main objective of this work was to explore the potential use of different physico-chemically characterized sugar beet by-products (pressed, ensiled, and dried pulp) as efficient and alternative sources of pectin following its extraction by acid or enzymatic methods. Likewise, the potential of the extracted pectins as thickening or gelling agents is investigated through their rheological characterization.

## 2. Results and Discussion

### 2.1. Overall Characterization of Sugar Beet By-Products

The results of the physicochemical analysis of the sugar beet by-products, reported in Table 1, show a reduction of °Brix, pH, dry weight (DW), protein, total carbohydrates, reducing carbohydrates, total dietary fiber, insoluble dietary fiber, and Mg when comparing the chemical composition of the sugar beet pulp pressed (SBP-P), with the sugar beet pulp ensiled (SBP-E) (*p* < 0.05).

This variation could be due to a fermentation process of DW (20–30%) during the storage time (seven–eight months) (Figure 1) carried out by saccharolytic and proteolytic bacteria; in a similar way to that observed by Álvarez et al. [14] who studied the effect of fermentation during silage of banana by-products.

Moreover, the SBP-E showed an increase in the amount of fat and SDF (*p* < 0.05). The higher fat and SDF content may be due to a greater cellular release, generated by the decrease in pH caused, in turn, as a result of the transformation of soluble sugars in acetic and lactic acid, as part of the metabolism of anaerobic bacteria [15]. On the other hand, the increase of Na and Fe (*p* < 0.05) found in SBP-E could be due to an increase in the solubility and bioavailability of minerals, as an effect of the pH reduction during silage [16].

Conversely, in the case of the sugar beet pulp dried (SBP-D), an expected reduction of a_w_, as well as a decrease in the content of reducing carbohydrates, soluble dietary fiber and antioxidant activity was observed, likely as a consequence of the heat treatment applied (Figure 1) (~100 °C/2–3 h) [17]. Lastly, a reduction of K was determined, probably due to a lower availability of this mineral for the analysis, as a consequence of the hardening of the sample by the drying effect (*p* < 0.05).

### 2.2. Pectin Extraction and Characterization

#### 2.1.1. Yield, Monomeric Composition and Protein

The application of the acid method allowed for the achievement of higher yields in comparison with the enzymatic method, regardless the type of sugar beet waste used (Table 2). This result is in line to that reported by Lim et al. [18] who compared the acid and the enzymatic method to extract pectin from Yuza (*Citrus junos*) pomace. The maximum yield was observed in the case of the pectin extracted from SBP-E followed by SBP-D using acid conditions (18.9 and 16.7%, respectively) (*p* < 0.05), which seems to be related with the high amount of soluble dietary fiber (SDF) observed in the ensiled and dried residue (Table 1). Despite yields obtained by the enzymatic method being lower than those observed with the acid method, the sugar beet ensiled residue (P-SBP-E-EM) gave rise to a higher yield (13.4%) in comparison with that reported by Zykwinska et al. [19] (4.0%) using a similar method of extraction, but instead, starting from fresh sugar beet pulp as raw material.

The analysis by GC-FID of the extracted pectins revealed the presence of xylose, arabinose, rhamnose, galactose, and galacturonic acid, whereas glucose could be derived from the acid hydrolysis of cellulose [20], by disruption of β-1,4-glycosidic bonds [21], and mannose from mannans and galactomannans [22,23] (Table 2). The acid extraction led to high quantities of xylose and arabinose in all cases, whereas the content in galacturonic acid (GalA) was significantly less important, which could be attributed to its acid degradation [24]. The enzymatic method, instead allowed obtaining pectin with a higher amount of galacturonic acid (GalA), and a lower amount of xylose and arabinose (*p* < 0.05).

However, it is important to notice that GalA was present in all extracted pectins in the range from 23.5% to 67.0%, having the pectins of sugar beet pulp ensiled (P-SBP-E-EM) and pressed (P-SBP-P-EM), both extracted by the enzymatic method, the highest GalA content (67.0 and 65.5%, respectively). In fact, these values suggest that pectin extracted from these by-products could be considered as food additives, according to the recommendations given by a Joint FAO/WHO Expert Committee on Food Additives, which established that pectin should not contain less than 65% of GalA calculated on the ash-free and dried basis [25].

The protein content was higher in the case of the pectin extracted by the acid method compared to the pectin extracted by the enzymatic method, possibly due to the severity of the acid method, resulting in a greater amount of protein residues linked to the pectin obtained. The pectin sample with the higher amount of protein was the pectin of sugar beet pulp pressed (P-SBP-P-AM), which is consistent with the greater amount of protein observed in the pressed by-product (SBP-P) (Table 1).

#### 2.1.2. Degree of Methoxylation (DM) and Molecular Weight (Mw)

FTIR spectra of pectins extracted from SBP-E by either acid or enzymatic methods are shown in Figure 2. The peak between 1052 and 1141 cm^−1^ is assigned to C=C double pectin bond. The absorption peaks at 1388 and 1633 cm^−1^ are related to the stretch bands of the pectin COO groups. These results indicate that the final products are true pectin compounds [26].

Pectins extracted by the enzymatic method regardless of the type of sugar beet pulp by-product (that is, pressed, silaged or dried) showed larger peaks at 1608 cm^−1^ and 1745 cm^−1^ than those observed at the same wavelengths for pectins extracted by the acid method. However, when the degree of methoxylation (DM) was calculated by correlating the peak area of the esterified carboxyl groups to the peak area of total carboxyl groups, the DM values of pectin extracted by the acid method were statistically similar to the DM of pectins extracted by the enzymatic method (*p* < 0.05) (Table 3).

DM values were in the range from 45.2 to 50.1%, which correspond to low-methoxyl pectins (DM < 50%). This type of pectins is known as “slow-gelling” and has the ability to form gels at slightly neutral or basic pH, with maximum consistency in the presence of calcium at concentrations ranging from 20 to 100 mg per gram of pectin, and/or with low amounts of sugar [27]. Therefore, low-methoxyl pectins are suitable as additives for the development of low-fat products, or foods by diabetic people [28].

Table 3 shows the molecular weight (Mw) of the extracted pectins estimated by SEC. This parameter was also statistically similar between the pectins extracted from sugar beet by-products pressed, silage, or dried by acid or enzymatic methods (*p* < 0.05). The determined Mw values were in the range from 303 to 322 kDa, in agreement to the maximum Mw values reported by Zykwinska et al. (2008) [19] for pectin extracted from fresh SBP (310 kDa).

#### 2.1.3. Emulsifying Activity Index (EAI)

Figure 3 shows the emulsifying activity index (*EAI*) of the different pectins extracted by acid or enzymatic methods from sugar beet pulp pressed, ensiled, or dried, using the commercial citrus pectin as standard. In general, the *EAI* of sugar beet pectins extracted by the acid method were higher than those of the *EAI* of sugar beet pectins extracted by the enzymatic method. Furthermore, all pectin samples extracted from sugar beet residues, excluding P-SBP-P-EM, showed a significant EAI higher than that of citrus pectin (*p* <0.05), according to the observed by Lerouxet al. [7].

Pectin from sugar beet pulp pressed extracted by the acid method (P-SBP-P-AM) and pectin of sugar beet pulp dried extracted by the same method (P-SBP-D-AM), presented the highest *EAI* (73.51, 75.53 m^2^/g, respectively) among all the assayed pectins (*p* < 0.05). These values are in the range reported by Huang et al. [29] (75.3–104.9 m^2^/g) for sugar beet pectin extracted from pulp dried. Remarkably, these authors pointed out that the studied pectin samples exhibited good emulsifying activity.

The difference in the emulsifying activity observed among the analysed pectins can be associated with the different content in protein (Table 2), which plays a predominant role in the emulsifying properties of sugar beet pectin [30]. Thus, it can be observed that pectins with higher *EAI* (P-SBP-P-AM and P-SBP-D-AM) had the highest protein content (4.3 and 4.1 g/100 g DW) (Table 2) and vice versa (*p* < 0.05). In fact, the emulsifying effect is related to the ability of protein molecules to open in aqueous-lipid media, allowing that their electrically charged outer groups to bind with water molecules, and the internal non-polar amino acids to be released and bound to the oily particles, linking both phases, until forming the stable mixture, called emulsion [31].

#### 2.1.4. Zeta Potential (ζ) and Apparent Viscosity (η)

The zeta potential (ζ) of sugar beet pectins extracted by acid or enzymatic methods from pressed, ensiled, and dried residues is presented in Figure 4. The isoelectric point was not reached in any of the samples, but it should occur at very acidic pHs, 1.5–2.0. Within the range of pH from 4.5 to 9, pectins extracted by the acid method had slightly higher absolute values of ζ (−25 to −34 mV) than pectins extracted by the enzymatic method (−20 to −28 mV) (*p* < 0.05). This indicates that pectin particles extracted by acid method showed higher stability in aqueous dispersion than those obtained by enzymatic method. This behavior is associated with the fact that acidic extraction increases the electronegativity of pectin, which causes the particles to move away from each other and remain suspended in the aqueous medium [32].

Citrus pectin exhibited an intermediate stability between the samples extracted by the enzymatic and the acid method, and lastly, the P-SBP-P-MA showed the highest stability in aqueous solution among all analyzed samples (*p* < 0.05).

Figure 5 shows the apparent viscosity (η) of pectin solutions prepared in water (20 mg/mL). The solutions prepared with all types of sugar beet pectin showed lower viscosity than solutions prepared with commercial citrus pectin; in turn, solutions prepared with sugar beet pectin extracted by the enzymatic method presented higher apparent viscosity values than solutions prepared with sugar beet pectin extracted by the acid method, regardless of the type of by-product used (*p* < 0.05). The higher viscosity observed in pectin extracted by the enzymatic method, could be explained by the presence of polyelectrolytes, since they affect the conformation of the macromolecule and the nature of the counterions, which act as a brake on the flow of polymers [33]. In this sense, the pectin extracted from sugar beet pulp ensiled by the enzymatic method (P-SBP-E-EM) reached the highest final viscosity (40 m Pa.s), followed by pectin of sugar beet pulp dried extracted by the enzymatic method (P-SBP-D-EM) (18 m Pa.s), while the pectin of sugar beet pulp dried, extracted by the acid method (P-SBP-D-AM), exhibited the lowest apparent viscosity value (4 m Pa.s).

The lower viscosity observed in the solutions prepared with pectin obtained from the dried beet residue may be due to the drying processes that can negatively affect the properties of the rheological properties of gum [29]. However, the viscosity of the pectin obtained in the present study, by the enzymatic method from the dry sugar beet by-product (P-SBP-D-EM) (18 m Pa.s), obtained in the industry by application of 140 °C/2–3 h with boiler gases (Figure 1), was higher than that reported by Huang et al. [29] for a sugar beet pectin obtained by the acid method (12 M HCl), from dehydrated pulp at 40, 50, and 60 °C/8h, in a hot air oven (10 m Pa.s); reaffirming the advantage of using the enzymatic method in the extraction process of pectin, in order to obtain high viscosity, since in both cases the waste used presented a moisture content close to 4.6% (~96% DW) Table 1.

It should be noted that, the degree of methoxylation (DM) and the molecular weight (Mw) of the extracted pectin (Table 3), could not have influenced the viscosity observed in the present study, since both the DM and the Mw did not have a statistically significant difference in all extracted pectins (*p* < 0.05).

Overall, the apparent viscosity of the pectin solutions decreased when the shear rate increased, which is indicative of a pseudoplastic (shear-thinning) flow behavior due to a decrease of entanglements of their structure, as is the case of gums [34].

## 3. Materials and Methods

### 3.1. Samples

Commercial citrus pectin was purchased from Acofarma (Barcelona, Spain). Industrial sugar beet by-products were provided by Azucarera Ebro (Madrid, Spain). Figure 1 shows the industrial process of sugar extraction and derived by-products. Briefly, sugar beet is washed, disinfected with hot water and grinded to obtain small particles named cossettes. Then, sugar is extracted from cossettes by a diffusion process with water heated at 70 °C. Water and sucrose are concentrated and dried, and the cossettes with a 7%–8% of dry weight (DW) are pressed to extract more sugar, obtaining the Sugar Beet Pulp Pressed (SBP-P), with a DW of 28–29%. This residue is stored in silos during 7–8 months obtaining the Sugar Beet Pulp Ensiled (SBP-E), and then, it can be dried by sun (3 days), or with boiler gas caldera 2–3 h until 88-96 % DW obtaining the Sugar Beet Pulp Dried (SBP-D). Those residues are destined to the direct sell or used in the production of animal feed. SBP-P, SBP-E and SBP-D were selected performing a simple, non-stratified random sampling. The beet used in the extraction process came from different cultivars of *Beta vulgaris* var. altissima Döll, harvested in early-January, in Spain. All samples were lyophilized, ground, sieved thought 250 µm mesh, and maintained at −20 °C until analysis.

### 3.2. Physicochemical Characterization of Sugar Beet by-Products

°Brix, pH, water activity (a_w_), dry weight (DW) and protein content were determined according to the AOAC methods described by Megías-Pérez, Gamboa-Santos, Soria, Villamiel, and Montilla, (2014) [35]. Fat content was determined by the soxhlet method using propanol during 2 h of heating. Minerals content was determined in the Interdepartmental Research Service (SIdI-UAM) (Madrid, Spain), by ICP-MS in an Elan 6000 Perkin-Elmer Sciex instrument (Concord, Canada).

Total carbohydrates were determined according to the phenol sulfuric method described by Masuko et al. [36]. Working inside a fume hood, 278 µL of aqueous dilutions of the samples (70 µg/mL) were disposed in Eppendorf tubes of 2 mL, and 167 µL of phenol sulfuric solution (5% *w/v*) were added on the dilutions. Tubes were stirred in a vortex, 1 mL of sulfuric acid were carefully added, and the mixture was shaken again, and then kept for 30 min without agitation. Afterwards, the absorbance was measurement at 480 nm in a Synergy HT Multi-Mode Microplate reader (BioTek^®^ Instruments, Inc., Winooski, VT, 05404-0998 USA) (Gen 5 software) and using a calibration curve of galacturonic acid (0–0.2 mg/mL). Results were expressed as the total carbohydrates (g/100 g DW).

Reducing carbohydrates were measured using the method described by Sumner et al. [37], by adding 100 µL of 3,5-dinitrosalicylic acid (DNS) reactive to 100 µL of the diluted sample previously located in the Eppendorf tubes of 1.5 mL. The mixture was stirred and boiled during 5 min, and then cooled in an ice bath, and 750 µL of milli-Q water was added. After shaking again, 280 µL of mix were transferred to a multiwell plate and the absorbance was measured at 540 nm. The calibration curve was prepared with GalA (0–0.4 mg/mL), and data were expressed as reducing carbohydrates (g/100 g DW).

Fiber content was determined by the enzymatic-gravimetric method described by McCleary et al. [38]. Samples were milled and sieved through 250 μm mesh. A dilution of 1 g of sample in 50 mL of sodium phosphate buffer 0.08 M pH 6 was prepared, and 100 µL of α-amylase from hog pancreas (Sigma-Aldrich Química SL, Madrid, Spain, ≥5000 U/mL) was added to remove the starch, heating at 95 °C for 15 min in a water bath with agitation. After cooling down, the pH was adjusted to 7.5 with 0.275M NaOH, and 5 mg of protease from *Streptomyces griseus* (Sigma Aldrich, ≥3500 U/g) was added and the mixture was heated (60 °C/ 30 min) to remove the protein.

The pH was adjusted to 4–4.6 with HCl 0.325 N, and 300 µL of amyloglucosidase from *Aspergillus niger* (Sigma Aldrich, 72,500 U/g) (1 mg/mL) (30 min, 60 °C) were added to remove the gelatinized starch. Afterwards, 280 mL of water at 60 °C were added and left to stand for 1 h to precipitate the insoluble fiber. The precipitate was filtered through a porous crucible of 0.8 µ, washed successively with 60 mL of ethanol 78%, 20 mL of ethanol 95%, and 20 mL of acetone. The solid detained was dehydrated for 24 h at 100 °C, and its final weight was corrected depending on the protein and ash value, to obtain the insoluble dietary fiber (IDF) content. Total dietary fiber (TDF) was determined by replacing the 280 mL of water by ethanol 95% and filtering all the precipitate; and soluble dietary fiber (SDT) was calculated by subtracting IDF from TDF values.

Total phenolic content was determined in methanolic extracts of samples by the Folin-Ciocalteu method described by Soria et al. [39]. To obtain the extracts, 0.2 g of powder sample were homogenized in 5 mL of methanol using an Ultra Turrax (IKA Labortechnik, Janke and Kunkel, Staufen, Germany) at 24,000 rpm for 1 min. The homogenates were placed in tubes of 15 mL and stirred at 750 rpm (50 °C/20 min) in an Eppendorf ThermoMixer^®^ incubator (15 mL). Mixtures were centrifuged at 2000× *g* for 15 min and filtered through Acrodisc PVDF syringe filters (0.45 μm, Sigma-Aldrich).

The reaction was carried out, by adding 100 μL of MeOH and 100 μL of Folin-Ciocalteu 2N to 100 μL of the filtered extract, disposed in Eppendorf tubes of 1.5 mL. After 5 min, 700 µL of Na_2_CO_3_ (75 g/L) were added, and tubes were left in the dark for 20 min. Mixtures were centrifuged at 28,000× *g* for 3 min, and the absorbance was measured in the supernatant at 735 nm using a Synergy HT Multi-Mode Microplate reader (BioTek^®^ Instruments, Inc., Winooski, VT 05404-0998, USA) (Gen 5 software). The calibration curve was prepared with gallic acid (0–60 mg/L) and the results were expressed as mg of gallic acid equivalent (GAE)/g DW.

Antioxidant capacity was determined according to the method proposed by Brand-Williams et al. [40], by the addition of 193 µL of 2,2-diphenyl-1-picrylhydrazyl (DPPH) 2 mM diluted in methanol (1:15) to 7 µL of methanolic extract of powder sample, in the Eppendorf tubes of 1.5 mL. Mixture was stirred and transferred to a multiwell cell (280 µL). After 30 min without agitation under dark conditions, the absorbance was measured at 517 nm. The calibration curve was prepared with Trolox (Sigma 648471, 500 mg; ≥98%) (0.25–2.5 mM in methanol). Results were expressed as mM Trolox/100 g DW.

### 3.3. Pectin Extraction

#### 3.3.1. Acid Method

Pectin was extracted by the traditional acidifying method optimized by Neha Babbar et al. [41] with slight modifications. The sample was mixed with deionized water (5%, *w/v*) and the pH was adjusted to 1.2 with HNO_3_ 12 M. The suspended samples were heated at 90 °C with continuous stirring at 200 rpm for 3 h. After the reaction was completed, the resulting slurries were cooled down to 40 °C, and the pH was adjusted to 4.5 with NH_3_.H_2_O 25% and centrifuged at 2600× *g* at 4 °C for 10 min, to separate insoluble fiber, protein, and other non-pectin compounds. The supernatant was collected and stored in a refrigerator at 4 °C for subsequent purification. One volume of supernatant was precipitated using two volumes of ethanol 95% for 1 h at room temperature. The centrifugation was repeated and the precipitate was washed three times with ethanol at 70%. After purification, the pectin was dried by lyophilization, and stored until its analysis.

#### 3.3.2. Enzymatic Method

According to the method described by Liew et al. [42], pectin was extracted from sugar beet by-products by dilution of powder samples in buffer sodium citrate 0.05 M at pH 4.5 (1:20 *w/v*) and heating with continuous stirring (125 rpm) with the commercial cellulase Celluclast^®^, derived from *Trichoderma reesei* (Novozymes Corp., Bagsvaerd, Denmark. 700 U/g) (1.17 U/g powder sample) at 61 °C during 102 min. Mixtures were left without stirring at room temperature during 24 h, to degrade the cellulose; and then, they were centrifuged at 2600× *g* at 4 °C for 10 min, to separate insoluble fiber, protein, and other non-pectin compounds. Ethanol 95% was added to the supernatants (2:1 *v/v*), and ethanolic mixtures were kept under dark conditions at 4 °C for 24 h to allow the flotation of pectin. Pectin solutions were centrifuged at 3400× *g* by 15 min and the precipitate was washed twice with ethanol 70%, mixed and centrifuged before each addition. Finally, pectin was de-colorated by adding acetone drop-by-drop and dried through lyophilization. The pectin yield was calculated by means of Equation (1):(1)$$\mathrm{Pectin}\;\mathrm{yield}\;\left(\%\right)=\frac{\mathrm{Weight}\;\mathrm{of}\;\mathrm{product}\;\mathrm{obtained}\;\left(g\right)}{\mathrm{Weight}\;\mathrm{of}\;\mathrm{powder}\;\mathrm{sample}\;\left(g\right)}\;\times \;100$$

### 3.4. Pectin Characterization

#### 3.4.1. Monomeric Composition

Sample was hydrolyzed with trifluoroacetic acid (TFA) 2 M (30 mg/1.5 mL) at 110 °C during 4 h [43]. Then, 500 µL of hydrolysate were placed in a flask and evaporated under vacuum at 43 °C. 400 µL of phenyl-β-d-glucoside (0.5 mg/mL) (internal standard, I.S.) were added, and the flask was evaporated again. For the oximes formation, 250 µL of hydroxylamine chloride in pyridine (2.5%) were added and the mixture was vortexed and heated at 70 °C during 30 min, stirring the sample at the beginning, at the middle, and at the final of the 30 min. Samples were persilylated with 250 µL of hexamethyldisylazane (HMDS) and 25 µL of TFA at 50 °C for 30 min, agited, and centrifuged at 10,000× *g* for 2 min.

The released monomers were analyzed by GC-FID (Agilent Technologies 7890A gas chromatograph, Agilent Technologies, Wilmington, DE, USA) using a DB-5HT capillary column (15 m × 0.32 mm × 0.10 µm) (J&W Scientific, Folsom, CA, USA). Injector and detector temperatures were 280 and 350 °C, respectively; oven temperature program was increasing from 150 °C to 165 °C at 1 °C/min and up to 300 °C at a heating rate of 10 °C/min. Nitrogen was used as the carrier gas, at a flow of 1 mL/min, and injections were made in split mode 1:20. Data acquisition was done using a HPChem Station software (Hewlett-Packard, Palo Alto, CA, USA). The response factors were calculated after the analysis of standard solutions (xylose, arabinose, rhamnose, galactose, mannose, glucose, and galacturonic acid), in concentrations of 0.01–2 mg, and 0.2 mg of I.S.

#### 3.4.2. Protein Content

Protein content was determined in all the pectin samples following the Bradford assay [44] using the Bio-Rad protein assay kit, which includes Coomasie Blue and bovine serum albumin (BSA) (0–2 mg/mL) for the calibration curve. The absorbances were measured at 595 nm and protein content was expressed as g/100 g DW.

#### 3.4.3. Degree of Methylesterification (DM)

The degree of methylesterification (DM) of extracted pectin was determined by Fourier transform infrared spectroscopy (FTIR) analysis. KBr discs were prepared mixing the pectin with KBr (1:100) and pressed. FTIR spectra Bruker IFS66v (Bruker Optics, Ettlingen, 76275 Germany) were collected at absorbance mode in the frequency range of 400–4000 cm^−1^, at a resolution of 4 cm^−1^ (mid infrared region) with 250 coadded scans. The DM was expressed as the ratio between the peak area of methylesterified carboxyl groups: COOCH_3_, measured at 1745 cm^−1^; and the sum of the peak areas of esterified carboxyl groups: COOCH_3_ at 1745 and free carboxyl groups COO^−^ measured at 1608 cm^−1^; according to the Equation described by Singthong et al. [45], Equation (2):(2)$$\mathrm{DM}=\frac{\mathrm{Methylesterified}\;\mathrm{carboxyl}\;\mathrm{groups}}{\mathrm{Total}\;\mathrm{carboxyl}\;\mathrm{groups}}\;\times \;100$$

#### 3.4.4. Molecular Weight (Mw)

The distribution of Mw of pectin samples was determined by Size Exclusion Chromatography (SEC) according to the method described by Muñoz-Almagro et al. [46] with slight modifications. Dilutions of sample in milli-Q water (1 mg/mL) were eluted with ammonium acetate 0.01 M at a flow rate of 0.5 mL/min for 50 min at 30 °C. The eluent was monitored using a refractive index detector (Boeblingen, Germain) at 30 °C, disposed in a LC Agilent Technologies 1220 Infinity LC System 1260 (Agilent Technologies, Boeblingen, Germain), equipped with two consecutive TSK-GEL columns (G5000 PWXL, 7.8 × 300 mm, particle size 10 μm, and G2500 PWXL, 7.8 × 300 mm, particle size 6 μm; Tosoh Bioscience, Stuttgart, Germany). Calibration curves were prepared using pullulans of Mw 788, 473, 212, 100, 1.3, and 0.34 kDa; and, Mw values were the average weight at peak maximum obtained in the analysis by triplicate.

#### 3.4.5. Emulsifying Activity

Emulsifying activity was calculated by turbidity according to the method described by Wang et al. [47] with slight modifications.

A volume of 100 mL of pectin solution in water (20%, *w/v*), were mixed with 5 g of corn oil using an Ultraturrax at 24,000 rpm for 1 min to obtain an emulsion. The emulsion was diluted 30, 500 and 900-folds with sodium dodecyl sulphate (SDS) (1 g/L). Turbidity of emulsions was measured in a UV spectrophotometer SPECORD^®^210 and the WinASPECT^®^ PLUS software (Analitik Jena AG, Jena, Germany), at 500 nm, using the SDS solution as the blank sample. The turbidity was calculated by Equation (3):(3)$$T=\frac{2.303\;\times \;A\;\times \;F}{I}$$
where T is turbidity of emulsions (m^−1^), A is the absorbance at 500 nm, F is the dilution factor, and I is path length, which is 0.01 m.

The emulsion activity index (*EAI*) was calculated using Equation (4):(4)$$EAI=\frac{2\;\times \;T}{\emptyset \;\times \;c}$$
where Ø is the oil volume fraction of the dispersed phase, and c is the concentration of pectin in the emulsion.

#### 3.4.6. Zeta potential (ζ)

Zeta potential (ζ) of pectin in aqueous dilution was determined according to the method described by Falk et al. [48], using a Malvern Zeta sizer Nano ZS instrument (Malvern Instruments Ltd., Worcestershire, UK). A volume of 250 mL of pectin solution was prepared by dissolving the extracted pectin in KCl 0.1 M (1 mg/mL). The solution was agitated, sonicated during 1 min, and its pH was measured (mixture 1). Then, 10 mL of mixture 1 and 90 mL of KCl 0.1 M were agitated, sonicated during 1 min, and its pH was recorded (mixture 2). Briefly, mixture 2 was injected into the clear disposable zeta cell and the ζ was measured. The procedure for the preparation of mixtures was repeated, in order to obtain dilutions of mixture 1, at different pH values (2 to 10) by adding HCl 0.1 M or KOH 0.1 M drops, and their respective ζ values were measured. The measuring cell was carefully washed after each reading, using deionized water and the next dilution, avoiding bubbles inside to evade measurement errors.

#### 3.4.7. Apparent Viscosity

Following the method described by Huang et al. [29], extracted pectin was dissolved in deionized water (20 mg/mL) using a magnetic stirrer at ambient temperature during 1 h. The apparent viscosity of the sample was determined using a Modular Advanced Rheometer System (MARS) (Thermo Fisher Scientific Inc., Waltham, MA, USA). Flow curves over the shear rate (1–100 s^−1^) were measured at 25 °C. The measuring geometry used was a double-cone and plate system with a truncated cone with an angle of 2° and a diameter pf 60 mm. The apparent viscosity and steady shear rate measurement were fitted to the Herschel-Bulkley model, Equation (5):(5)$$\sigma =\sigma_{0}+k\;\times \;{\overset{•}{\Upsilon }}^{n}$$
where σ is the shear stress (Pa), $\sigma_{0}$ is the yield stress (Pa), k is the consistency index (Pa.s^n^), $\overset{•}{\Upsilon }$ is shear rate (s^−1^), and n is the flow behavior index.

### 3.5. Statistical Analysis

Extractions and analysis were carried out at least in triplicate and means were compared by Tukey’s test (*p* < 0.05), using SPSS Statistics 22.0 (IBM Corp., Armonk, NY, USA). Differences were expressed as mean ± standard deviation.

## 4. Conclusions

The present study compared the compositional and rheological properties of sugar beet pectin, which was efficiently extracted from pressed, ensiled, and dried residues by acid or enzymatic methods. The silage process caused a reduction in the protein and insoluble carbohydrates content of sugar beet pulp, as well as an increase in the fat and soluble dietary fiber amount, likely due to a lactic fermentation process. The drying process, instead, caused a reduction in the reducing carbohydrates, soluble fiber and antioxidant capacity. Either the type of sugar beet by-product or the extraction method had no impact on the degree of methoxylation and molecular weight of extracted pectin. Nevertheless, the enzymatic method allowed the extraction of pectin with a significantly higher content of galacturonic acid as compared to the acid method, due to the milder conditions of the former. The rheological analysis showed that all pectins obtained presented a pseudoplastic flow behavior. Furthermore, the zeta potential and *EAI* values indicated that pectins extracted by the acid method showed good stabilizing behavior in aqueous dispersion and good emulsifying activity, whereas pectins enzymatically extracted had a higher apparent viscosity that was linked to the presence of polyelectrolytes that impede the polymer flow.

To conclude, the information provided in the present work could be very useful for the potential reuse of ensiled and dried by-products from sugar beet industry in the cost-effective production of pectin with different technological properties depending on the applied extraction method. Pectins were conveniently characterized and with suitable rheological properties are known to find immediate applications in the pharmaceutical and/or food fields.

## Figures and Tables

**Figure 1 molecules-24-00392-f001:**
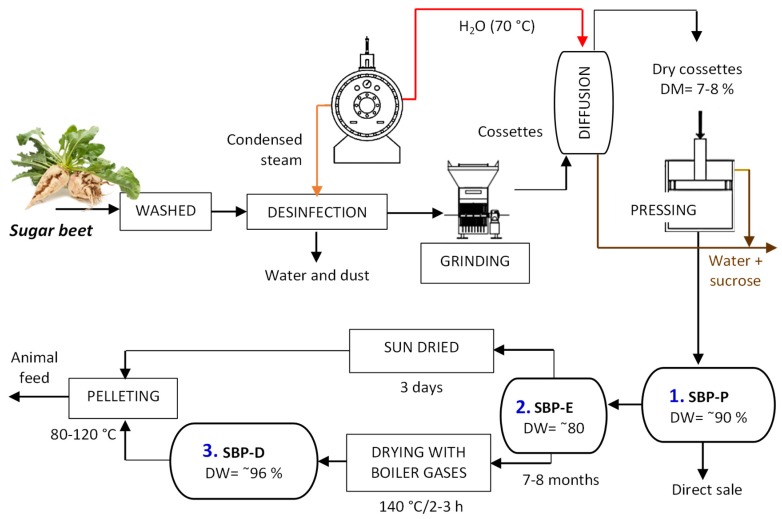
Industrial process of sugar extraction form sugar beet. By-products obtained: 1. SBP-P: sugar beet pulp pressed, 2. SBP-E: sugar beet pulp ensiled, 3. SBP-D: sugar beet pulp dried. DW: dry weight.

**Figure 2 molecules-24-00392-f002:**
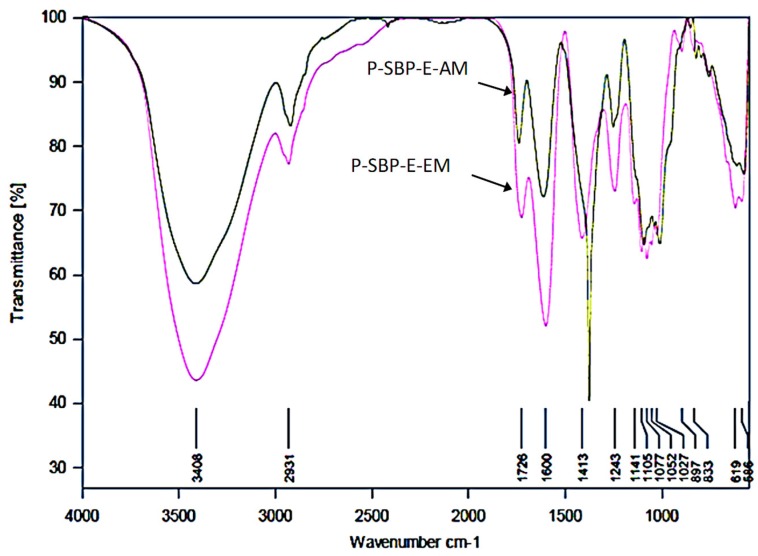
FTIR spectra of sugar beet pulp pectin. P-SBP-E-AM: pectin of sugar beet pulp ensiled, extracted by acid method. P-SBP-E-EM: pectin of sugar beet pulp ensiled, extracted by enzymatic method.

**Figure 3 molecules-24-00392-f003:**
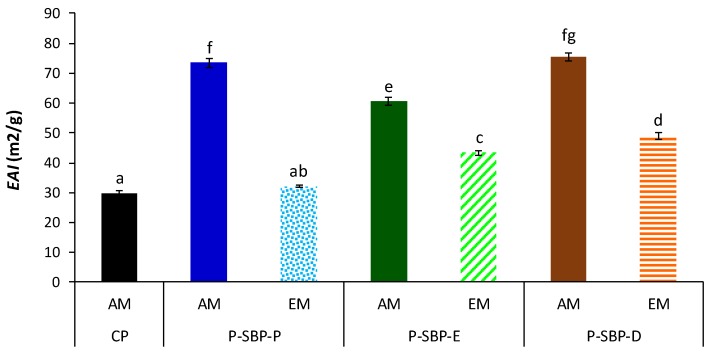
Emulsifying activity index *(EAI)* of pectin obtained from sugar beet by-products by acid or enzymatic methods. CP: citrus pectin. P-SBP-P: pectin of sugar beet pulp pressed, P-SBP-E: pectin of sugar beet pulp ensiled, P-SBP-D: pectin of sugar beet pulp dried. AM: Acid method. EM: Enzymatic method. Different letters (a–d) in the columns denote significant difference (*p* < 0.05).

**Figure 4 molecules-24-00392-f004:**
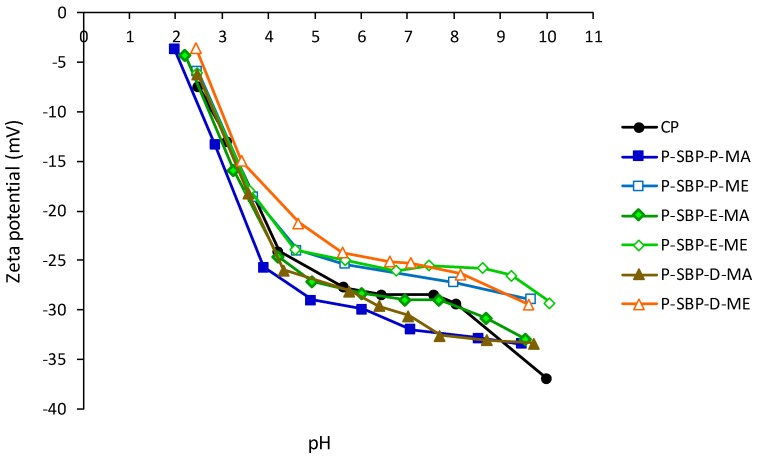
Zeta potential (ζ) curves of pectin. CP: citrus pectin. P-SBP-P-AM: pectin from sugar beet pulp pressed, acid method; P-SBP-P-EM: pectin from sugar beet pulp pressed, enzymatic method; P-SBP-E-AM: pectin from sugar beet pulp ensiled, acid method; P-SBP-E-EM: pectin from sugar beet pulp ensiled, enzymatic method; P-SBP-D-AM: pectin from sugar beet pulp dried, acid method; P-SBP-D-EM: pectin from sugar beet pulp dried, enzymatic method.

**Figure 5 molecules-24-00392-f005:**
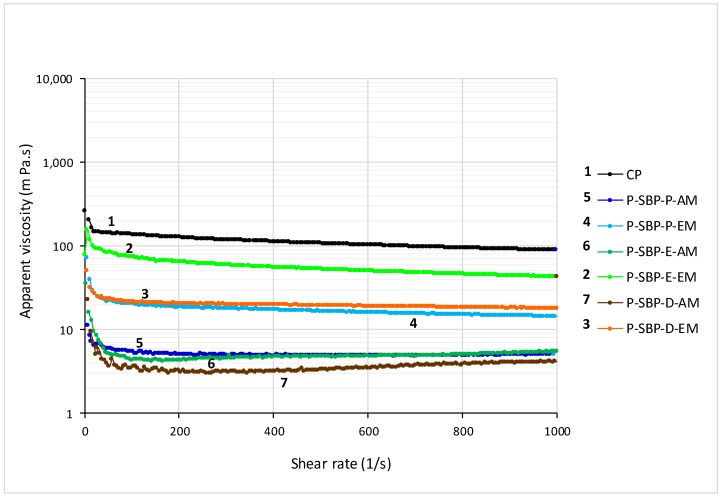
Apparent viscosity of pectin solutions (20 mg/mL). CP: citrus pectin. P-SBP-P-AM: pectin of sugar beet pulp pressed, acid method. P-SBP-P-EM: pectin of sugar beet pulp pressed, enzymatic method. P-SBP-E-AM: pectin of sugar beet pulp ensiled, acid method. P-SBP-E-EM: pectin of sugar beet pulp ensiled, enzymatic method. P-SBP-D-AM: pectin of sugar beet pulp dried, acid method. P-SBP-D-EM: pectin of sugar beet pulp dried, enzymatic method.

**Table 1 molecules-24-00392-t001:** Chemical composition of sugar beet pulp by-products (g/100 g DW).

Parameter	SBP-P	SBP-E	SBP-D
°Brix	5.00 ± 0.21 ^b^	4.60 ± 0.07 ^a^	4.40 ± 0.14 ^a^
pH	4.62 ± 0.08 ^c^	3.51 ± 0.03 ^a^	3.71 ± 0.06 ^a,b^
Aw	0.88 ± 0.01 ^b^	0.90 ± 0.02 ^b^	0.73 ± 0.02 ^a^
DW (%)	91.12 ± 0.16 ^b^	83.41 ± 0.15 ^a^	96.53 ± 0.22 ^c^
Total fat (g/100 g DW)	0.84 ± 0.02 ^a^	1.70 ± 0.04 ^c^	1.33 ± 0.03 ^b^
Protein (g/100 g DW)	10.42 ± 0.52 ^c^	8.30 ± 0.30 ^ab^	8.01 ± 0.31 ^a^
Total carbohydrates (g/100 g DW)	82.64 ± 0.63 ^c^	70.22 ± 0.32 ^a^	78.14 ± 0.50 ^b^
Reducing carbohydrates (g/100 g DW)	10.40 ± 0.11 ^c^	6.73 ± 0.10 ^b^	4.21 ± 0.08 ^a^
TDF (g/100 g DW)	75.20 ± 0.24 ^b^	64.52 ± 0.07 ^a^	76.84 ± 0.32 ^b,c^
IDF (g/100 g DW)	47.58 ± 0.16 ^b^	34.51 ± 0.09 ^a^	51.32 ± 0.18 ^b,c^
SDF (g/100 g DW)	26.63 ± 0.47 ^a^	30.04 ± 0.50 ^b,c^	29.78 ± 0.34 ^b^
Ash (g/100 g DW)	1.86 ^a,b^	1.87 ^a,b^	1.81 ^a^
Na^+^ (mg/100 g DW)	17.66 ^a,b^	26.39 ^c^	16.16 ^a^
Mg^+2^ (mg/100 g DW)	244.97 ^c^	217.16 ^a,b^	214.30 ^a^
P^+3^ (mg/100 g DW)	27.51 ^a,b^	24.20 ^a^	23.55 ^a^
K^+^ (mg/100 g DW)	184.72 ^b^	189.84 ^b,c^	174.68 ^a^
Ca^+2^ (mg/100 g DW)	1327.90 ^a,b^	1332.21 ^b,c^	1317.26 ^a^
Fe^+3^ (mg/100 g DW)	54.63 ^a^	84.95 ^c^	62.45 ^a,b^
Total phenols (mg GAE/100 g DW)	0.38 ± 0.05 ^c^	0.29 ± 0.02 ^b^	0.17 ± 0.01 ^a^
Antioxidant capacity (mM de Trolox/100 g DW)	2.34 ± 0.14 ^b,c^	2.23 ± 0.09 ^b^	1.16 ± 0.10 ^a^

SBP-P: sugar beet pulp pressed, SBP-E: sugar beet pulp ensiled, SBP-D: sugar beet pulp dried. FM: fresh matter. DW: dry weight. TDF: total dietary fiber. IDF: insoluble dietary fiber. SDF: soluble dietary fiber. Means with different letters a–c denote significant difference (*p* < 0.05) in the same row.

**Table 2 molecules-24-00392-t002:** Yield extraction (g pectin/100 g DW), monomeric composition and protein of pectin from sugar beet by-products (g/100 g DW).

Pectin	Extraction Method	Yield	Xylose	Arabinose	Rhamnose	Galactose	Galacturonic Acid	Mannose	Glucose	Protein
P-SBP-P	AM	13.60	33.35 ± 1.11 ^f^	3.60 ± 0.08 ^e,f^	9.81 ± 0.31 ^e^	14.50 ± 0.50 ^d^	23.52 ± 0.11 ^a^	6.14 ± 0.18 ^e^	2.01 ± 0.02 ^b^	4.3 ± 0.22 ^e,f^
	EM	3.91	6.36 ± 0.19 ^c^	0.10 ± 0.00 ^a^	3.20 ± 0.06 ^a^	8.26 ± 0.27 ^a,b^	65.51 ± 0.30 ^e^	4.04 ± 0.03 ^c^	5.22 ± 0.09 ^d^	1.6 ± 0.06 ^a^
P-SBP-E	AM	18.94	25.48 ± 0.69 ^d,e^	3.22 ± 0.06 ^e^	4.94 ± 0.13 ^c,d^	7.60 ± 0.28 ^a^	42.74 ± 0.23 ^b^	2.39 ± 0.04 ^a^	3.30 ± 0.03 ^c^	3.4 ± 0.14 ^d^
	EM	13.40	4.53 ± 0.14 ^a^	0.70 ± 0.01 ^b^	3.50 ± 0.11 ^a,b^	9.68 ± 0.33 ^c^	66.98 ± 0.80 ^ef^	5.05 ± 0.15 ^d^	0.76 ± 0.01 ^a^	2.0 ± 0.08 ^b^
P-SBP-D	AM	16.72	21.53 ± 0.85 ^d^	2.82 ± 0.06 ^d^	4.74 ± 0.15 ^c^	8.88 ± 0.24 ^b^	48.92 ± 0.43 ^d^	4.75 ± 0.17 ^d^	0.62 ± 0.01 ^a^	4.1 ± 0.20 ^e^
	EM	7.50	5.22 ± 0.16 ^a,b^	0.79 ± 0.02 ^b,c^	16.79 ± 0.69 ^f^	8.68 ± 0.26 ^b^	44.80 ± 0.37 ^b,c^	3.06 ± 0.09 ^b^	12.57 ± 0.30 ^e^	2.8 ± 0.1 ^c^

DW: dry weight. P-SBP-P: pectin from sugar beet pulp pressed, P-SBP-E: pectin from sugar beet pulp ensiled, P-SBP-D: pectin from sugar beet pulp dried. AM: Acid method. EM: Enzymatic method. Means with different letters a–f denote significant difference (*p* < 0.05) in the same column.

**Table 3 molecules-24-00392-t003:** Degree of methoxylation (DM) (%), and molecular weight (Mw)(kDa) of pectin extracted from sugar beet by-products.

Pectin	Extraction Method	DM	* Mw
P-SBP-P	AM	49.29 ^a^	306 ± 7 ^a^
	EM	47.08 ^a^	311 ± 9 ^a^
P-SBP-E	AM	50.14 ^a^	303 ± 7 ^a^
	EM	48.36 ^a^	322 ± 10 ^a^
P-SBP-D	AM	48.39 ^a^	315 ± 9 ^a^
	EM	45.21 ^a^	319 ± 10 ^a^

P-SBP-P: pectin of sugar beet pulp pressed, P-SBP-E: pectin of sugar beet pulp ensiled, P-SBP-D: pectin of sugar beet pulp dried. AM: Acid method. EM: Enzymatic method. Means with similar letter (a) in the same column do not present significant difference (*p* < 0.05). * Mw was calculated as the average at peak maximum observed in the analysis followed by triplicate.

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
