# Peer review of "Structural and Rheological Properties of Pectins Extracted from Industrial Sugar Beet By-Products"

_molecules, 2019, doi:10.3390/molecules24030392_

Round 1
Reviewer 1 Report
The manuscript is written in a fairly well written English, still has some structures from Spanish that could be improved by a native English speaker. My may concern would be on the treatment and discussion of the data. It seems that data from table 2 and 3 are not fully exploited on sections 2.2.3, and 2.2.4; for they are far better than references 33 and 34. On the other hand figure 1 would be of use better without equipment images and more simple but informative description to enhance the conditions involved in the different stages of the process that are eventually related to the differences found in the pectins.
Finally, FT-IR, has very good information not mentioned in the text, like the region within 1052 and 1141 for molecular identity, to mention one of many.
Author Response
Point 1. The manuscript is written in a fairly well written English, still has some structures from Spanish that could be improved by a native English speaker. My may concern would be on the treatment and discussion of the data. It seems that data from table 2 and 3 are not fully exploited on sections 2.2.3, and 2.2.4; for they are far better than references 33 and 34. On the other hand figure 1 would be of use better without equipment images and more simple but informative description to enhance the conditions involved in the different stages of the process that are eventually related to the differences found in the pectins.
Response 1. The language has been revised as requested.
The data discussion of sections 2.2.3 and 2.2.4 has been expanded and related in the manner requested with Tables 2 and 3, respectively (Lines 178-183 and 192-194, 229-239).
Figure 1 has been improved and all available information about the sugar extraction process and sugar beet by-product treatment has been described.
Point 2. Finally, FT-IR, has very good information not mentioned in the text, like the region within 1052 and 1141 for molecular identity, to mention one of many.
Response 2. The interpretation on FT-IR, has been extended indicating the most important for the case of pectins, that is, the description of peaks at 1052-1141 cm-1 and the absorption peak at 1388-1633 cm-1 (Lines 126-128).
Thanks for the observation!

Reviewer 2 Report
General comments to manuscript number molecules-417690:This manuscript entitled “structural and rheological properties of pectins extracted from industrial sugar beet by-products” by Teresa Pacheco et al describes the yield, viscosity and sugar composition of pectins from sugar beet by-products by specific extraction and spectroscopic methods.However the manuscript needs some revision before being acceptable. The specific comments below should help the authors to improve the quality of the manuscript.
Specific comments:
Abstract:
Line 13: What were the types of enzymes?
Introduction:
Line 28-33: Could you provide references on numbers of hectares given ?
Line 46: The percentage of acetyl group in sugar beet pectin was higher than 4-5%:
Physical characterisation of the rhamnogalacturonan and homogalacturonan fractions of sugar beet (Beta vulgaris) pectin. Gordon A. Morrisa, Marie-Christine Ralet, Estelle Bonnin. Jean-François Thibault, Stephen E. Harding, Carbohydrate Polymers 82 (2010) 1161–1167 :
DA 35%
Characterisation of pectins extracted from fresh sugar beet under different conditions using an experimental design, Sebastien Levigne, Marie Christine Ralet, Jean Francois Thibault, Carbohydrate Polymers 49 (2002 ) 145-153
DA 58%
Results and discussion
Confusion was present throughout the manuscript between DM: dry matter and DM: degree of methyl esterification. In order to solve this problem, a suggestion:
Dry matter = dry weight (DW)
Degree of methyl esterification = DM
Table 1: it is better to give the charge of all the ions.
Line 69: Have studies been conducted to detect the bacteria present?
Figure 1: explain DM in legends.
Line 99-100: No glucuronic acid was detected in pectin extract?
Figure 2: Explain the peak at 1410 cm-1 in P-SBP-E-AM. Why it is not present in the P-SBP-E-EM?
Is it possible to determine the Degree of acetylation (DA) with this FTIR method?
Could you give the DA of all the samples studies?
Line 135: DE? Is it DM?
Figure 3: The contrast between the two green colors is not good. Could you change this?
Figure 4: The contrast between the two green colors is not good. Could you change this?
Figure 5: The contrast between the two green and two blue colors are not good. Could you change this?
Materials and methods:
Line 225: What is the method of disinfected?
Line 244: Could you provide references for the phenol sulfuric method?
Line 251: Could you provide references for the DNS method?
Line 262: “Streptomyces griseus” should be italicized.
Line 264: “Aspergillus niger” should be italicized.
Line 273: Could you provide references for the Folin-Ciocalteu method?
Line 286: Could you provide references for the antioxidant capacity method?
Line 290: What is the source of the “Trolox”?
Line 319: Remove the “%” at the end of the equation.
Line 349: Is it possible to determine the Degree of acetylation (DA) with this FTIR method?
Line 358: Remove the “%” at the end of the equation but add it after DM.
In order to calculate the DM, did you make a calibration curve with pectin with known DM?
Line 375: p/v = w/v?
Line 393: Could you provide references for the zeta-potential determination method?
References:
Check all the references carefully: they are not all in the same format, missing titles or volume numbers.
Author Response
This manuscript entitled “structural and rheological properties of pectins extracted from industrial sugar beet by-products” by Teresa Pacheco et al describes the yield, viscosity and sugar composition of pectins from sugar beet by-products by specific extraction and spectroscopic methods. However the manuscript needs some revision before being acceptable. The specific comments below should help the authors to improve the quality of the manuscript.
Specific comments:
Abstract:
Point 1. Line 13: What were the types of enzymes?
· Cellulase (Line 13)
· Celluclast® (Novozymes Corp., Bagsvaerd, Denmark. 700 U/g), that is a commercial cellulase product, derived from Trichoderma reesei (Lines 335-336).
Introduction:
Point 2. Line 28-33: Could you provide references on numbers of hectares given?
Response 2. The reference is:
(1). Eurostat Agriculture, forestry and fishery statistics; Forti, R., Ed.; 2017th ed.; Imprimerie Centrale: Luxembourg, 2017; ISBN 978-92-79-63350-8.
In the revised version this reference is quoted in lines 31 (following referee’s remark) and 36.
Point 3. Line 46: The percentage of acetyl group in sugar beet pectin was higher than 4-5%:
Physical characterisation of the rhamnogalacturonan and homogalacturonan fractions of sugar beet (Beta vulgaris) pectin. Gordon A. Morrisa, Marie-Christine Ralet, Estelle Bonnin. Jean-François Thibault, Stephen E. Harding, Carbohydrate Polymers 82 (2010) 1161–1167:
DA 35%
Characterisation of pectins extracted from fresh sugar beet under different conditions using an experimental design, Sebastien Levigne, Marie Christine Ralet, Jean Francois Thibault, Carbohydrate Polymers 49 (2002) 145-153
DA 58%
Response 3. In our study we did not determine the degree of acetylation, but the degree of methyl esterification.
About sugar beet pectin properties in comparison with a citrus and apple pectin Leroux, Langendorff, Schick, Vaishnav & Mazoyer (2003), point out that:
"Sugar beet pectin (SBP), compared to the main sources of pectin which are citrus and apple, have poorer gelling properties due to their higher content of neutral sugars, low presence of acetyl groups (4-5%)" (Lines 45- 47)
But at the same time they indicate that:
"the degree of acetylation does not significantly affect the emulsifying capacity of pectins, unlike the protein content (Leroux, Langendorff, Schick, Vaishnav & Mazoyer, 2003)"
Therefore, to clarify the description, "and/or" has been added in the mentioned paragraph (Lines 45-48).
Results and discussion
Point 4. Confusion was present throughout the manuscript between DM: dry matter and DM: degree of methyl esterification. In order to solve this problem, a suggestion:
Dry matter = dry weight (DW)
Degree of methyl esterification = DM
Response 4. The change has been made throughout the document as the reviewer suggested. Thank you very much for the suggestion.
Point 5. Table 1: it is better to give the charge of all the ions.
Response 5. The charge of the ions has been specified as requested (Table 1).
Point 6. Line 69: Have studies been conducted to detect the bacteria present?
Response 6.The type or quantity of bacteria present in the sugar beet byproducts has not been determined, because the study does not contemplate the consumption of the by-products directly. The objective was their use to obtain pectin by means of methods that, in addition to low pH, involve high temperature, therefore, no contamination in the final product was expected. However, microbiological determination in the byproducts and pectins obtained, could be addressed in the future, in another study, focused on assessing the safety of consumption of these pectins, analyzing the effect of the industrial process and the extraction method on the microbiological content.
Point 7. Figure 1: explain DM in legends.
Response 7.DM has changed to "DW: dry weight" in the Figure 1 and in the legend (Line 88).
Point 8. Line 99-100: No glucuronic acid was detected in pectin extract?
Response 8.The determination of glucuronic acid has not been accomplished, since galacturonic acid is the main component of pectins, and its determination and quantification is sufficient to classify it as a food additive according to the Joint FAO/WHO Expert Committee on Food Additives (FAO, & WHO, 2009). (Lines 112-116).
Point 9. Figure 2: Explain the peak at 1410 cm-1 in P-SBP-E-AM. Why it is not present in the P-SBP-E-EM?
Is it possible to determine the Degree of acetylation (DA) with this FTIR method?
Could you give the DA of all the samples studies?
Response 9. The peak at 1410 in the case of the pectin extracted by the acid method (Figure 2) may be due to the presence of aldehyde groups with torsion in the C-H groups (FQUIM – UNAM, 2018), which could happen due to the electronegativity, as a result of the low pH.
The degree of acetylation can be carried out by gas-liquid chromatography, high performance liquid chromatography, enzymatic, titrimetric or colorimetric methods after derivation (Cochrane, 1975; Voragen, Schols, & Pilnik, 1986; Krop et al., 1974, MacComb and MacCready, 1957, Schultz, 1962, Wood and Siddiqui, 1971; Levigne, Thomas, Ralet, Quemener & Thibault, 2002).
The DA of the obtained pectins has not been reported in the present study, since this parameter has less influence than the protein on the emulsifying properties, as mentioned in previous lines.
To understand the usefulness of DM and DA it is necessary to consider the structure of pectin. It mainly consists of a GalA -rich backbone, known as homogalacturonan (HG ≈ 65%), of which a number of residues are methyl esterified at the C-6 position, thereby conferring a specific degree of methoxyl esterification (DM) to the polymer. This degree of esterification and its distribution pattern define the charge distribution over the polymer playing a major role in the dimerization of pectin chains through the formation of junction zones, either via cooperative Ca2+ complexation or at reduced water activity as well as pH, thus defining the gelation properties of pectin.
We determined the degree of methoxylation, since it is one of the main molecular parameters to be considered for the technological properties at the industry. The functional properties of pectins in foods and their reactivity toward calcium and other cations are largely dependent on the amount of unmethylated GalA subunits and their distribution pattern within the HG stretches. In fact, pectins are classified in Low Methoxyl Pectin (LMP) and High Methoxyl Pectin (HMP) taking into account this parameter (Luzio & Cameron, 2013).
Point 10. Line 135: DE? Is it DM?
Response 10. The correct is DM. Thank you!
(Line 140)
Point 11. Figure 3: The contrast between the two green colors is not good. Could you change this?
Response 11. The color and the filling of the bars has been improved (Figure 3, Line 163).
Point 12. Figure 4: The contrast between the two green colors is not good. Could you change this?
Response 12. The contrast between the colors has been modified as much as possible (Figure 4, Line 195). Thank you.
Point 13. Figure 5: The contrast between the two green and two blue colors are not good. Could you change this?
Response 13. That is the best possible contrast, and unfortunately, we cannot change the markers, since the large number of points (1000 data) prevents that they are observed. Nevertheless, in order to improve the clarity of the figure, we have included the numbers both in the lines and in the name of each series (pectin) (Figure 5, Line 219).
Materials and methods:
Point 14. Line 225: What is the method of disinfected?
Response 14.It is only a disinfection with hot water (Line 247-248) coming from the steam condensate of the boilers, as indicated in Figure 1. Then, the liquid destined to obtain sucrose (water + sucrose) is disinfected with ozone or SO2, but that stage is not included in the process of interest for our study since we focused on the by-products.
Furthermore, based on a remark from reviewer 1, Figure 1 has been improved and all available information about the sugar extraction process and sugar beet by-product treatment has been now fully described.
Point 15. Line 244: Could you provide references for the phenol sulfuric method?
Response 15. The information has been added (Lines 266-267).
Point 16. Line 251: Could you provide references for the DNS method?
Response 16. The change has been made (Line 274).
Point 17. Line 262: “Streptomyces griseus” should be italicized.
Response 17. The change has been made (Line 286).
Point 18. Line 264: “Aspergillus niger” should be italicized.
Response 18.The change has been made (Line 288).
Point 19. Line 273: Could you provide references for the Folin-Ciocalteu method?
Response 19.The information has been added (Line 298).
Point 20. Line 286: Could you provide references for the antioxidant capacity method?
Response 20. The information has been added (Line 311).
Point 21. Line 290: What is the source of the “Trolox”?
Response 21. Sigma 648471 - 500 mg; ≥98%. This information is now included in the revised version of the manuscript (Line 316).
Point 22. Line 319: Remove the “%” at the end of the equation.
Response 22. Removed symbol (Line 346).
Point 23. Line 349: Is it possible to determine the Degree of acetylation (DA) with this FTIR method?
Response 23. The determination of degree of acetylation of pectins, by means of chromatography, is the most robust method, as has been indicated in previous lines.
Point 24.Line 358: Remove the “%” at the end of the equation but add it after DM.
Response 24. Removed symbol (Line 386).
Point 25.In order to calculate the DM, did you make a calibration curve with pectin with known DM?
Response 25. In this kind of studies, a calibration curve is not necessary. The DM was expressed as the ratio between the peak area of methylesterified carboxyl groups: COOCH3, measured at 1745 cm−1; and the sum of the peak areas of esterified carboxyl groups: COOCH3 at 1745 and free carboxyl groups COO- measured at 1608 cm−1; according to the described by Singthong et al. (2004). There is a plethora of studies on DM of pectin that use the same technique.
Point 26.Line 375: p/v = w/v?
Response 26.Yes. Thank you for the correction and sorry for the mistake (Line 403)
Point 27.Line 393: Could you provide references for the zeta-potential determination method?
Response 27.The information has been added (Line 421-422).
References:
Point 28.Check all the references carefully: they are not all in the same format, missing titles or volume numbers.
Response 28.All citations and references have been reviewed and placed in the Vancouver format used by Molecules Journal.
Thanks for all the observations and recommendations.
References:
Cochrane, G. C. (1975). Journal of Chromatography Science, 13, 440-447.
FAO, & WHO. (2009). Compendium of food additive specifications. (Joint FAO/WHO Expert Committee on Food Additives, Ed.) (Vol. 4). Pages 1-119. Rome-Italy. http://www.fao.org/3/a-i0971e.pdf
Accessed on October 8, 2018.
FQUIM - UNAM. (2018). Regiones básicas de un espectro de infrarrojo. http://depa.fquim.unam.mx/amyd/archivero/TablasIR_15437.pdf
Accessed on Janury 10, 2019.
Leroux, J., Langendorff, V., Schick, G., Vaishnav, V., & Mazoyer, J. (2003). Emulsion stabilizing properties of pectin. Food Hydrocolloids, 17:455–462.
Levigne, S., Thomas, M., Ralet, M.-C., Quemener, B., & Thibault, J.-F. (2002). Determination of the degrees of methylation and acetylation of pectins using a C18 column and internal standards. Food Hydrocolloids, 16(2002), 547-550.
Luzio, G. A., & Cameron, R. G. (2013). Determination of degree of methylation of food pectins by chromatography. Journal of the Science of Food and Agriculture, 93(10), 2463–2469.
MacComb, E. A., & MacCready, R. M. (1957). Analytical Chemistry, 29, 819-821.
Schultz, T. H. (1962). In R. L. Whistler & J. N. BeMiller, Methods in carbohydrate chemistry (pp. 187-189). vol. 5. New York: Academic Press.
Singthong, J.; Cui, S.W.; Ningsanond, S.; Goff, H.D. Structural characterization, degree of esterification and some gelling properties of Krueo Ma Noy (Cissampelos pareira) pectin. Carbohydr. Polym. 2004, 58, 391–400.
Voragen, A. G. J., Schols, H. A., & Pilnik, W. (1986). Determination of the degree of methylation and acetylation of pectins by H.P.L.C. Food Hydrocolloids, 1(1), 65-70.
Wood, P. J., & Siddiqui, I. R. (1971). Analytical Biochemistry, 39, 418-428.

Round 2
